# LMIC-PRIEST: Derivation and validation of a clinical severity score for acutely ill adults with suspected COVID-19 in a middle-income setting

Carl Marincowitz[1]*, Peter Hodkinson[2], David McAlpine[2], Gordon Fuller[1], Steve Goodacre[1], Peter A. Bath[1,3], Laura Sbaffi[3], Madina Hasan[1], Yasein Omer[3], Lee Wallis[3]

1 Centre for Urgent and Emergency Care Research (CURE), Health Services Research School of Health and Related Research, University of Sheffield, Sheffield, United Kingdom, 2 Division of Emergency Medicine, Groote Schuur Hospital, Observatory, University of Cape Town, Cape Town, South Africa, 3 Information School, University of Sheffield, Sheffield, United Kingdom

* c.marincowitz@sheffield.ac.uk

**Data Availability Statement:** The data used for this study are subject to a data sharing agreement with the Western Cape Government Department of

## Abstract

### Background

Uneven vaccination and less resilient health care systems mean hospitals in LMICs are at risk of being overwhelmed during periods of increased COVID-19 infection. Risk-scores proposed for rapid triage of need for admission from the emergency department (ED) have been developed in higher-income settings during initial waves of the pandemic.

### Methods

Routinely collected data for public hospitals in the Western Cape, South Africa from the 27th August 2020 to 11th March 2022 were used to derive a cohort of 446,084 ED patients with suspected COVID-19. The primary outcome was death or ICU admission at 30 days. The cohort was divided into derivation and Omicron variant validation sets. We developed the LMIC-PRIEST score based on the coefficients from multivariable analysis in the derivation cohort and existing triage practices. We externally validated accuracy in the Omicron period and a UK cohort.

### Results

We analysed 305,564 derivation, 140,520 Omicron and 12,610 UK validation cases. Over 100 events per predictor parameter were modelled. Multivariable analyses identified eight predictor variables retained across models. We used these findings and clinical judgement to develop a score based on South African Triage Early Warning Scores and also included age, sex, oxygen saturation, inspired oxygen, diabetes and heart disease. The LMIC-PRIEST score achieved C-statistics: 0.82 (95% CI: 0.82 to 0.83) development cohort; 0.79 (95% CI: 0.78 to 0.80) Omicron cohort; and 0.79 (95% CI: 0.79 to 0.80) UK cohort. Differences in prevalence of outcomes led to imperfect calibration in external validation. However,

Health and Wellness, which prohibits further sharing of patient-level data. Access to these and related data should be requested directly from this organization and is subject to the necessary ethical and organizational approval processes. Please contact Health.Research@westerncape.gov.za to request these data.

**Funding:** CM is a National Institute for Health Research (NIHR) Clinical Lecturer in Emergency Medicine (Grant Number Not Applicable/NA). This work is part of the Grand Challenges ICODA pilot initiative, delivered by Health Data Research UK and funded by the Bill & Melinda Gates Foundation and the Minderoo Foundation. The Provincial Health Data Centre (PHDC), Health Intelligence Directorate, Western Cape Government Health and Wellness acknowledges funding from the United States National Institutes of Health (R01HD080465, U01AI069911), Bill and Melinda Gates Foundation (1164272; 1191327; INV-004657, INV-017293), the Wellcome Trust (203135/Z/16/Z), the United States Agency for International Development (72067418CA00023). The funders had no role in study design, data collection and analysis, decision to publish, or preparation of the manuscript.

**Competing interests:** The authors have declared that no competing interests exist.

use of the score at thresholds of three or less would allow identification of very low-risk patients (NPV $\geq$0.99) who could be rapidly discharged using information collected at initial assessment.

## Conclusion

The LMIC-PRIEST score shows good discrimination and high sensitivity at lower thresholds and can be used to rapidly identify low-risk patients in LMIC ED settings.

## Background

The severity of illness associated with COVID-19 has been reduced by mass vaccination and emergence of less severe variants. However, emergency health care systems in low- and middle- income countries (LMIC) may still be at risk of being overwhelmed during periods of increased infection, due to uneven vaccination and less resilient health care systems [1, 2]. Risk-stratification scores including the UK Royal College of Physicians National Early Warning Score, version 2 (NEWS2) and the COVID-specific Pandemic Respiratory Infection Emergency System Triage (PRIEST) score have been proposed to aid clinical decision-making around need for inpatient admission in the Emergency Department (ED) in patients with suspected COVID-19 [3–5]. Such risk-stratification scores were developed in high-income settings during initial waves of the pandemic [3, 4]. Validation of these scores in the middle-income setting of the Western Cape, South Africa, demonstrated good discrimination [6]. However, the scores did not outperform existing clinical decision-making and used predictors and physiological cut-offs that are not part of routine clinical practice in this setting.

In LMICs, disposition decision-making is based on clinician experience and gestalt [7]. Use of risk-stratification scores to allow rapid triage of need for hospitalisation can help prevent hospitals being overwhelmed and assist less-experienced clinicians during periods of increased COVID infection. To be applicable and easily useable in LMICs, a risk-stratification score must be based upon existing clinical triage practice. In South Africa patient acuity on arrival to the ED is triaged using the South African Triage Scale (SATS) [8]. The Western Cape of South Africa presented a unique opportunity to use routinely collected linked electronic health-care data, in a setting with a high degree of COVID case ascertainment compared to similar settings [9], to develop a risk-stratification score applicable to LMICs.

Our study aimed to:

1. Develop a contextually appropriate clinical severity score for patients with suspected COVID-19 in an Emergency Department setting in the Western Cape.

2. Externally validate the developed score.

## Methods

### Study design

This observational cohort study used routinely collected clinical data from EDs across the Western Cape, from the Hospital Emergency Centre Triage and Information System (HEC-TIS) [10] data repository, to develop a clinical risk-stratification score for ED patients with suspected COVID.

The performance of the risk-stratification score was externally validated in patients who presented during the Omicron wave and in a cohort of patients from the UK Pandemic

Respiratory Infection Emergency System Triage (PRIEST) study (collected during the first wave) [11].

The study was conducted and reported in accordance with Transparent reporting of a multi-variable prediction model for individual prognosis or diagnosis (TRIPOD) and Reporting of studies Conducted using Observational Routinely-collected Data (RECORD) guidelines [12, 13].

## Setting

For the development and Omicron period validation cohorts, data were collected from patients with suspected COVID-19 infection who attended public-sector EDs in the Western Cape Province. This is one of nine provinces in South Africa, and has almost 7 million inhabitants, of whom three quarters use public sector services [14]. A convenience sample of seven hospital EDs (based on use of the HECTIS system) was selected, representing predominantly urban, Cape Town metropole district and, a large peri-rural hospital ED. Clinical decision-making was largely based on clinician experience, contextualised to the local status: at times hospitals were overwhelmed and admission thresholds were raised [15, 16]. No specific prognostic score were applied in the ED beyond routine triage with SATS.

The external validation population was derived from the PRIEST mixed prospective and retrospective cohort study that collected data from 70 EDs across 53 sites in the UK between 26th March and 28th May 2020 [3].

## Data sources and linkage

In the Western Cape, data on ED clinical presentation are routinely collected by the HECTIS system, including presenting complaint, triage variables and outcome of consultation. Through deterministic matching, based on unique patient hospital numbers (performed by the Western Cape Provincial Health Data Centre (PHDC)) [14], linked data were obtained which included COVID test results from the National Health Laboratory Services (NHLS), comorbidities (based on prior health system encounters), data around admissions and movements within the health care system during the index COVID encounter, and death (if within, or reported to, the health care system). For patients with multiple ED attendances, data were extracted for the first ED attendance and outcomes were assessed up to 30 days from index attendance.

Data collection for the UK PRIEST cohort study has been described in previous publications [3, 17]. An anonymised version of the study data was used to derive the external validation cohort [18].

## Inclusion criteria

Our Western Cape study sample consisted of all adults (aged 16 years and over) at the time of first (index) ED attendance between 27th August 2020 and 11th March 2022, where a clinical impression of suspected, or confirmed, COVID-19 infection had been recorded. For those with multiple presentations, analysis was limited to the index presentation. Patients who presented after the emergence of the Omicron variant (November 2021) were included in a validation cohort [19].

Patients for whom age or sex were not recorded were excluded from analyses.

## Outcome

The primary composite outcome, in the Western Cape population, was intubation or non-invasive ventilation in the ED on index attendance, Intensive Care Unit (ICU) admission or

inpatient death up to 30 days from index attendance. This was comparable to the PRIEST study primary outcome of death or organ support (respiratory, cardiovascular, or renal) by record review at 30 days.

The secondary outcomes were: 1) death and 2) ICU admission (organ support in UK cohort), up to 30 days.

## Patient characteristics and candidate predictor variables

Physiological parameters and presenting complaints at triage at index ED presentation were extracted from the HECTIS database. Where no comorbidities were recorded, they were assumed not to be present. Implausible physiological variables were set as missing, including systolic blood pressure <50 mm HG, temperature >42 or <25 degrees, heart rate < 10/minute, oxygen saturation < 10% and respiratory rate = 0/minute.

Candidate predictor variables were selected on the basis of a previous systematic review of factors suitable for use in LMICs, previous research and availability at ED triage in the Western Cape [1, 3, 20]. Variables included: age, sex, presenting symptoms (cough or fever), co-morbidities (heart disease, diabetes, HIV, chronic lung disease, hypertension or pregnancy), physiological parameters and supplemental oxygen. Asthma/COPD was excluded from analysis due to an implausible protective relationship identified in preliminary modelling. Oxygen saturations include those measured where supplemental oxygen was already being administered when patients were initially triaged in the ED.

## Prognostic model development

The model development cohort was randomly split into derivation and internal validation cohorts. Candidate predictors were combined in a multivariable regression with Least Absolute Shrinkage and Selection Operator (LASSO) using ten-sample cross-validation to select models. The LASSO began with a full model of candidate predictors and simultaneously performed predictor selection and penalisation during model development to avoid overfitting. The LASSO was performed twice: first, when the number of predictors were unrestricted, and a second time, with restriction to ten predictors. Estimates of selected model discrimination and calibration were performed in the split internal validation cohort.

Continuous variables were modelled using fractional polynomials to account for non-linear forms and using categories based on TEWS (Triage Early Warning Score) (S1 Table in S1 File) [8]. As TEWS categories are used as part of existing triage, unless alternative modelling methods demonstrated significant increases in accuracy, these categories were planned *a priori* to be used in the clinical severity score. Three multivariable analyses were completed using different approaches to missing predictor variable data in the derivation cohort for comparison: (1) Complete case; (2) Multiple imputation using chained equations (10 imputations); (3) Deterministic imputation with missing variable assumed to be in the normal range using TEWS categorisation.

## Clinical severity score derivation and validation

Clinical members of the research team reviewed the models and selected variables for inclusion in the triage score, based on the prognostic value and consistency of selection across models, the clinical credibility of their association with the primary outcome, and their availability in the South African ED setting. Selected variables were categorised and assigned integer values using TEWS and the PRIEST score, if present in these clinical scoring systems, whilst checking that categorisation reflected the relationship between the variable and adverse outcome in the derived models. Additionally, selected variables were assigned integer values to

each category of predictor variable, based on the coefficient derived from a multivariable logistic regression model using categorised continuous predictors. This generated a composite clinical score in which risk of adverse outcome increased with the total score.

We applied the clinical score to the model development cohort, the Western Cape Omicron period and UK PRIEST external validation cohorts, calculating diagnostic parameters at each threshold of the score, constructing a receiver-operating characteristic (ROC) curve, calculating the area under the ROC curve (C-statistic) and calculating the proportion with an adverse outcome at each level of the score. Calibration plots for the risk-score were estimated in the external validation cohorts. We used deterministic imputation to handle missing data in the validation cohort, assuming missing predictor variable data were within normal physiological categories but excluding cases with fewer than three predictor variables.

All analyses were completed in STATA version 17 [21].

## Sample size

The sample size was fixed based on a census sample of patients in the Western Cape recorded on the HECTIS during the study period. In the smallest prognostic model development cohort, there were 102, 503 patients with over 100 outcomes per predictor parameter.

## Ethics

Use of routinely collected electronic health care records from the Western Cape for the derivation of the development and Omicron cohorts for this study was approved by the University of Cape Town Human Research Ethics Committee (HREC 594/2021), and the Western Cape Health Research Committee (WC_202111_034). All data were full anonymised at source before being provided to the research team and need for patient consent was waived.

Data collection for the UK validation cohort was first approved by the North West—Haydock Research Ethics Committee on 25 June 2012 (reference 12/NW/0303) and on the updated PRIEST study on 23rd March 2020. The Confidentiality Advisory Group of the Health Research Authority granted approval to collect data without patient consent in line with Section 251 of the National Health Service Act 2006. An anonymised form of the dataset was used for analysis (available on reasonable request to the PRIEST research team).

## Patient and Public Involvement (PPI)

A community advisory board (CAB) comprising eight community members affected by COVID (infected themselves or immediate family infected/ hospitalised). PPI members were recruited by an experienced community liaison officer with links to key community groups. Members were intentionally sought to be representative of the various population groups and demographics of the population. Through several meetings, the CAB were kept abreast of the study, and given the opportunity to input on the outcomes, particularly the acceptability of the risk-stratification score.

# Results

## Study populations

Fig 1 and S2 Fig in S1 File summarise population selection for the study cohorts. Table 1 summarises the characteristics of the 305,564 patients used for model development. S3 & S4 Tables in S1 File present the characteristics of the 140,520 patients in the Omicron, and 20,698 patients in the UK, validation cohorts. In total, 12,610 patients (4.13%, 95% CI:4.06% to 4.2%) experienced the primary outcome in the development cohort. This compared to, 2,787 patients

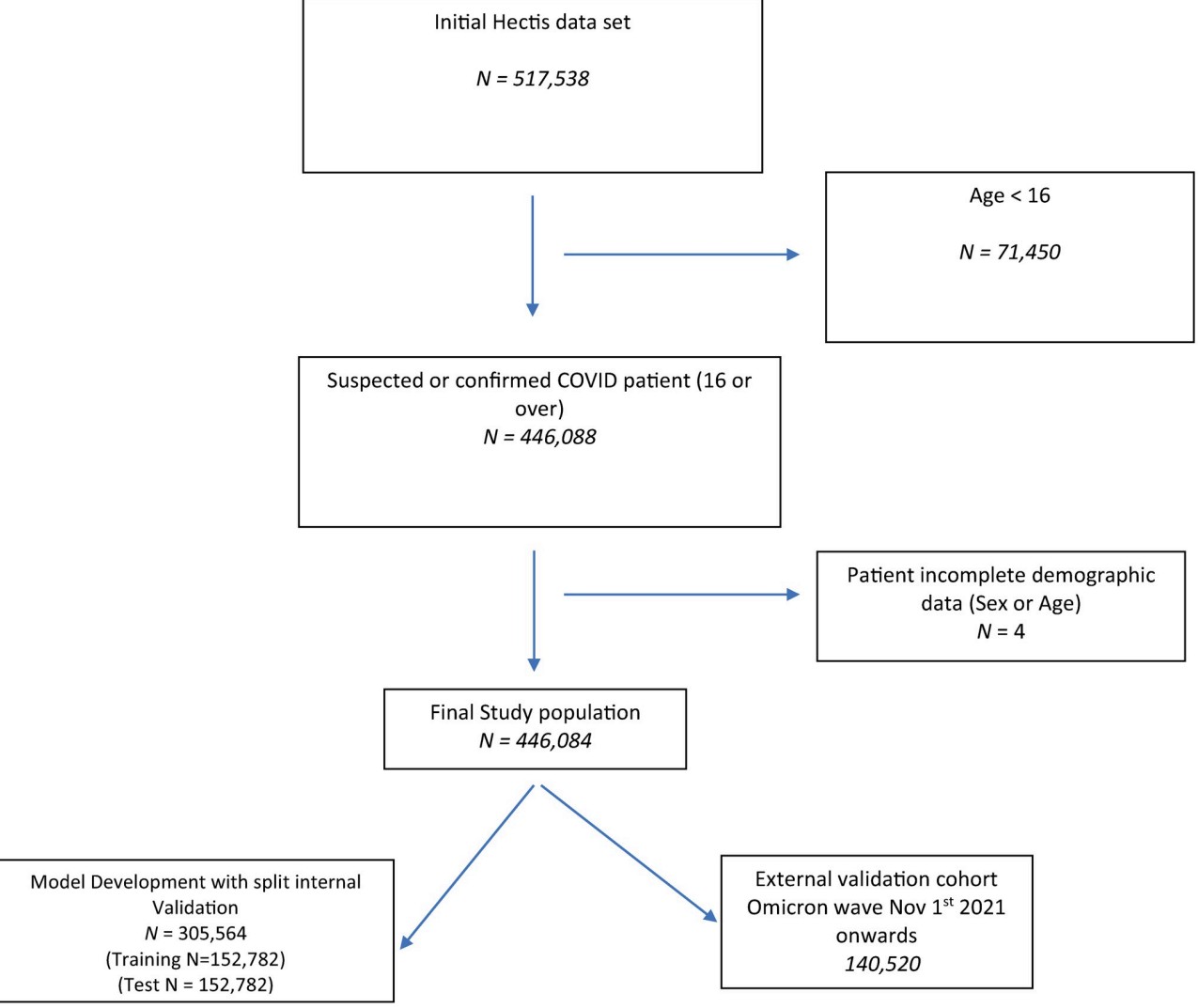

**Fig 1. Flow diagram of study population selection Western Cape, South Africa.**

(1.98%, 95% CI: 1.91% to 2.06%) in the Omicron period, and 4,579 patients (22.12%, 95% CI: 21.56% to 22.69%) in the UK, validation cohorts. In total, 74,580 patients used for model development (24.41%, 95% CI: 24.26 to 24.56%) had a diagnosis of COVID confirmed by PCR testing.

## Prognostic models

S5-S7 Tables in S1 File show the results of multivariable analysis restricted to inclusion of 10 predictor variables using complete case analysis, multiple imputation and deterministic imputation. S8-S10 Tables in S1 File show the results of the unrestricted multivariable analyses. S11, S12 Figs in S1 File present the corresponding calibration plots from split internal validation. Unrestricted LASSO on multiply imputed data with modelling of continuous variables using multifractional polynomials produced the model with the highest C-statistic (0.87, 95% CI 0.866 to 0.874) and calibration in the large (CITL) of -0.017 (95%CI -0.043 to 0.009). However,

**Table 1. Patient characteristics by outcome in the model development cohort.**

| Characteristic | Statistic/level | Adverse outcome | No adverse outcome | Total |
|---|---|---|---|---|
| | N | 12,610 (4.1%) | 292,954 (95.9%) | 305,564 |
| Age (years) | Mean (SD) | 56.5 (17.5) | 43.2 (17.1) | 43.7 (17.3) |
| | Median (IQR) | 59 (43, 70) | 40 (29, 56) | 41 (29, 57) |
| | Range | 16 to 105 | 16 to 110 | 16 to 110 |
| Sex | Male | 6,670 (52.9%) | 151,294 (51.6%) | 157,964 (51.7%) |
| | Female | 5,940 (47.1%) | 141,660 (48.4%) | 147,600 (48.3%) |
| Comorbidities | Asthma/COPD | 2,220 (17.6%) | 42,590 (14.5%) | 44,810 (14.7%) |
| | Other Chronic respiratory disease | 69 (0.6%) | 649 (0.2%) | 718 (0.2%) |
| | Diabetes | 5,256 (41.7%) | 51,622 (17.6%) | 56,878 (18.6%) |
| | Hypertension | 5,863 (46.5%) | 80,099 (27.3%) | 85,962 (28.1%) |
| | Immunosuppression (HIV) | 1,553 (12.3%) | 50,824 (17.4%) | 52,377 (17.1%) |
| | Heart Disease | 4,560 (36.2%) | 53,664 (18.3%) | 58,224 (19.1%) |
| | Pregnant | 62 (0.5%) | 1,915 (0.7%) | 1,977 (0.7%) |
| AVPU | Missing | | | 9,229 (3.0%) |
| | Alert | 9,159 (72.6%) | 264,460 (90.3%) | 273,619 (89.6%) |
| | Voice | 288 (2.3%) | 3,682 (1.3%) | 3,970 (1.3%) |
| | Confused | 617 (4.9%) | 11,661 (4%) | 12,278 (4%) |
| | Pain | 593 (4.7%) | 2,202 (0.8%) | 2,795 (0.9%) |
| | Unresponsive | 1,355 (10.8%) | 2,318 (0.8%) | 3,673 (1.2%) |
| Systolic BP (mmHg) | Missing | | | 10,389 (3.4%) |
| | N | 11,801 | 283,374 | 295,175 |
| | Mean (SD) | 130.9 (29.4) | 131.9 (25.5) | 131.8 (25.6) |
| | Median (IQR) | 128 (110,146) | 129 (115,145) | 129 (115,144) |
| | Range | 50 to 289 | 50 to 300 | 50 to 300 |
| Pulse rate (beats/min) | Missing | | | 9,995 (3.3%) |
| | N | 11, 858 | 283,711 | 295,569 |
| | Mean (SD) | 98.8 (23.4) | 93.5 (21) | 93.7 (21.1) |
| | Median (IQR) | 98 (83,113) | 92 (79, 106) | 92 (79,107) |
| | Range | 11 to 300 | 10 to 300 | 10 to 300 |
| Respiratory rate (breaths/min) | Missing | | | 9,969 (3.3%) |
| | N | 11,850 | 283,745 | 295,595 |
| | Mean (SD) | 22.2 (6.7) | 18.6 (4.1) | 18.8 (4.3) |
| | Median (IQR) | 20 (18,25) | 18 (16,20) | 18 (16,20) |
| | Range | 2 to 60 | 1 to 60 | 1 to 60 |
| Oxygen saturation | Missing | | | 27, 781 (6.2%) |
| | N | 11,634 | 274,409 | 286,043 |
| | Mean (SD) | 89.7 (12) | 96.2 (5.5) | 96 (6) |
| | Median (IQR) | 94 (86, 98) | 98 (96, 99) | 97 (95, 99) |
| | Range | 10 to 100 | 10 to 100 | 10 to 100 |
| Oxygen administration | Missing | | | 18,794 (6.2%) |
| | 1 (air) | 6,254 (49.6%) | 254,399 (86.8%) | 260,653 (85.3%) |
| | 2 (40% O2) | 346 (2.7%) | 5,360 (1.8%) | 5,706 (1.9%) |
| | 3 (28% O2) | 8 (0.1%) | 222 (0.1%) | 230 (0.1%) |
| | 4 (Nasal prongs) | 1,123 (8.9%) | 8,389 (2.9%) | 9,512 (3.1%) |
| | 5 (FM neb) | 27 (0.2%) | 571 (0.2%) | 588 (0.2%) |
| | 6 (rebreather mask) | 1,538 (12.2%) | 5,199 (1.8%) | 6,737 (2.2%) |
| | 7 (nasal prongs and rebreather mask) | 368 (2.9%) | 884 (0.3%) | 1,252 (0.4%) |
| | 8 intubated | 1,917 (15.2%) | 0 | 1,917 (0.6%) |
| | 9 NIV | 165 (1.3%) | 0 | 165 (0.1%) |

*(Continued)*

**Table 1.** (Continued)

| Characteristic | Statistic/level | Adverse outcome | No adverse outcome | Total |
|---|---|---|---|---|
| Temperature (°C) | Missing | | | 9,252 (3%) |
| | N | 12, 010 | 284,302 | 296,312 |
| | Mean (SD) | 36.4 (1.3) | 36.3 (0.8) | 36.4 (0.9) |
| | Median (IQR) | 36.4 (35.9, 37) | 36.3 (36, 36.7) | 36.3 (36, 36.7) |
| | Range | 25 to 41 | 25 to 42 | 25 to 42 |
| Cough | Missing | | | 41,524 (29.6%) |
| | Present | 557 (4.4%) | 8,538 (2.9%) | 9,095 (3%) |
| Fever | Missing | | | 93,962 (30.8%) |
| | Present | 178 (1.4%) | 2,829 (1%) | 3,007 (1%) |
| COVID PCR | Positive | 10,908 (86.5%) | 63,672 (21.7%) | 74,580 (24.4%) |
| Hospital admission | ICU | 1,527 (12.1%) | 0 | 1,527 (0.5%) |
| Death | Within 30 days contact | 9,711 (77%) | 0 | 9,711 (3.2%) |

restriction of modelling to 10 predictors and categorisation of continuous variables using TEWS only marginally reduced measures of accuracy in internal validation (worst performing restricted models; C-statistic 0.85 (95%CI 0.845 to 0.855) and CITL 0.126 (95% CI: 0.098 to 0.155)).

When restricted, there were eight predictors that were retained in all analyses (age, use of supplemental oxygen, oxygen saturation, diabetes, consciousness level, heart disease, respiratory rate, heart rate). Sex was retained in all but one complete case analysis.

## Clinical severity score derivation and validation

Clinical review judged that the eight predictors retained in all models and sex (included in all but one model) were clinically credible and should be included in the clinical severity score. As TEWS categories for physiological parameters are used clinically in the Western Cape and categorisation did not materially reduce measures of model accuracy, these were used in the risk-score. TEWS also routinely includes measurement of systolic blood pressure and temperature, which were retained in unrestricted models and therefore also included. The co-morbidities diabetes and heart disease were assigned scores based on the relative size of their coefficients across models. As age had a similar modelled form and effect size to the original UK PRIEST study, it was assigned categories and scores based on the PRIEST score. The developed score is shown in Table 2.

The LMIC-PRIEST score was applied to the model development cohort, Omicron and UK PRIEST validation cohorts. The estimated ROC curves for the primary outcome of these analyses are presented in Fig 2. S13, S14 Figs in S1 File show the estimated ROC curves when estimating the secondary outcome of 1) death or 2) admission to ICU/organ support. The score achieved better estimated discrimination when predicting death (C-statistic range: 0.79 UK cohort to 0.83 development cohort) compared to organ support/ICU admission (C-statistic range: 0.68 Omicron cohort to 0.74 development cohort). Fig 3 shows the calibration plots for performance of the score in the external validation cohorts. The score overestimated risk in the Omicron cohort as risk increased and systematically underestimated risk in the UK cohort.

Existing clinical decision-making to admit patients to hospital from the ED in the South African setting had a sensitivity of 0.77 (95% CI 0.76 to 0.78) and specificity of 0.88 (95% CI 0.87 to 0.88) for the primary outcome (prevalence primary outcome 3.45%). The positive predictive value (PPV) was 0.18 (95% CI 0.18 to 0.18) and the negative predictive value was

**Table 2. LMIC-PRIEST score (Score 0–27).**

| Variable | Range | Score |
|---|---|---|
| Respiratory rate (per minute) | 9–14 | 0 |
| | 15–20 | 1 |
| | <9 or 21–29 | 2 |
| | >29 | 3 |
| Oxygen saturation (%) | >95 | 0 |
| | 94–95 | 1 |
| | 92–93 | 2 |
| | <92 | 3 |
| Heart rate (per minute) | 51–100 | 0 |
| | 41–50 or 101–110 | 1 |
| | <41 or 111–129 | 2 |
| | >129 | 3 |
| Systolic BP (mmHg) | 101–199 | 0 |
| | 81–100 | 1 |
| | 71–80 or >199 | 2 |
| | <71 | 3 |
| Temperature (˚C) | 35–38.4 | 0 |
| | <35 or >38.4 | 2 |
| Alertness | Alert | 0 |
| | Reacts to voice | 1 |
| | Confused or reacts to pain | 2 |
| | Unresponsive | 3 |
| Inspired oxygen | Air | 0 |
| | Supplemental oxygen | 2 |
| Sex | Female | 0 |
| | Male | 1 |
| Age (years) | 16–49 | 0 |
| | 50–65 | 2 |
| | 66–80 | 3 |
| | >80 | 4 |
| Diabetes | No | 0 |
| | Yes | 2 |
| Heart disease | No | 0 |
| | Yes | 1 |

(NPV) 0.99 (95% CI 0.99 to 0.99). Clinicians discharged 85.28% of patients on first presentation. Table 3 presents the estimated sensitivity, specificity, negative and positive predictive values for levels of the score for the primary outcome in the three study populations that could be used clinically to inform admission decisions. S15-S17 Tables in S1 File shows these values for each level of the score and S18 Chart in S1 File shows the risk of the primary outcome for at each level in the 3 study cohorts.

## Discussion

### Summary

The LMIC-PRIEST score has been developed using a large cohort of patients with suspected COVID and a study period that encompasses Beta, Delta and Omicron waves in the Western

i) Model development cohort (N=282,051) C-stat 0.825 (95% CI: 0.821 to 0.828)

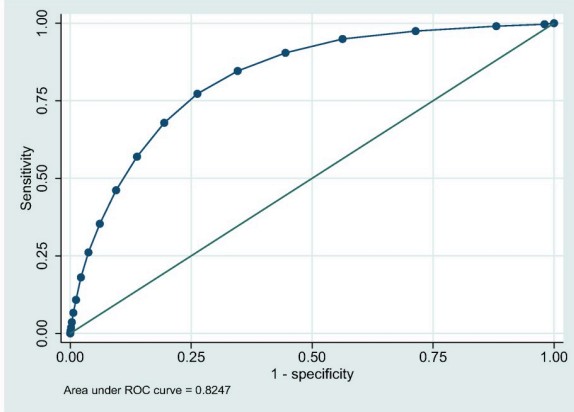

ii) Omicron Validation cohort (N=130,407) C-stat 0.792 (95% CI: 0.784 to 0.799)

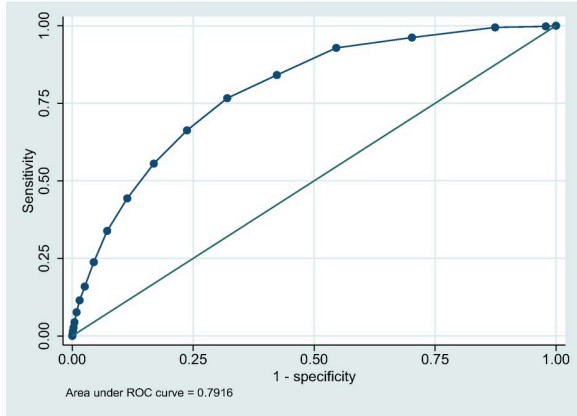

iii) UK PRIEST Validation cohort (N=17,669) C-stat 0.792 (95% CI: 0.786 to 0.799)

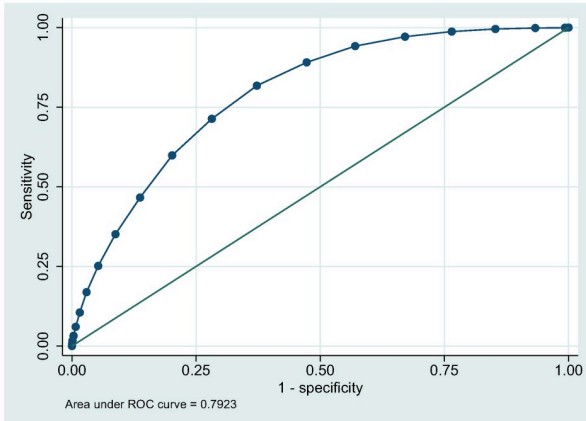

**Fig 2. ROC curves for predicting primary outcome for LMIC-PRIEST score.**

Cape. Alongside use of an external validation cohort, we were able to assess accuracy in both different income settings and variants. The LMIC-PRIEST score has shown consistent discrimination across different settings, C-statistics: 0.82 (95% CI: 0.82 to 0.83) development cohort, 0.79 (95% CI: 0.78 to 0.80) Omicron cohort and 0.79 (95% CI: 0.79 to 0.80) UK

i) Omicron Validation cohort (N=130,407)

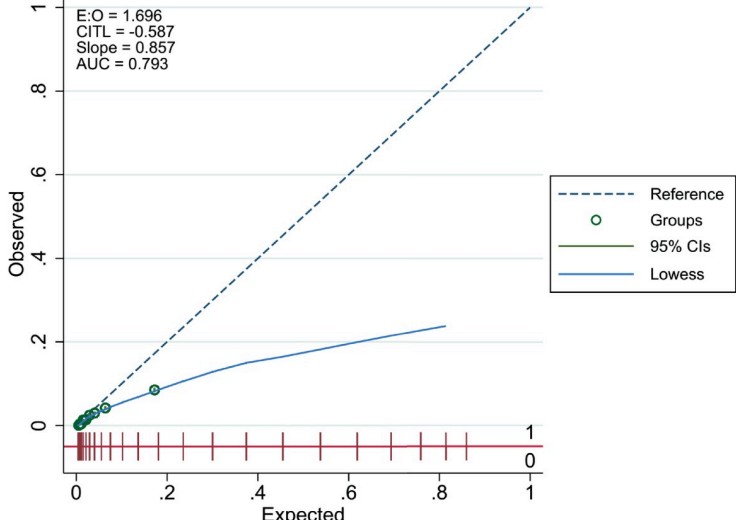

ii) UK PRIEST Validation cohort (N=17,669)

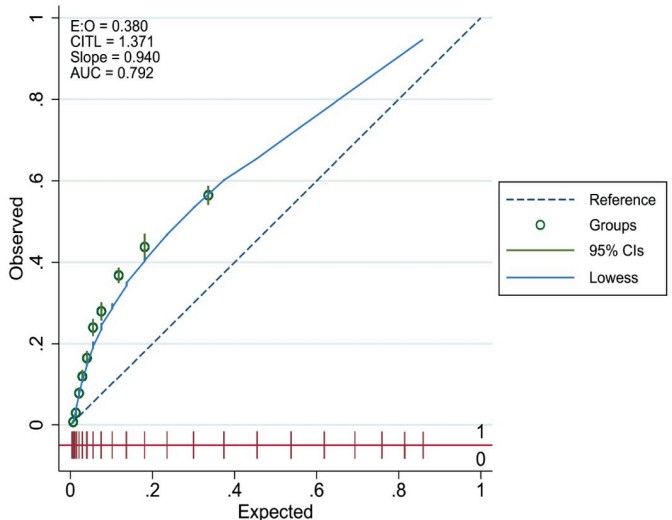

**Fig 3. Calibration curves for LMIC-PRIEST score performance in external validation.**

validation cohort. However, differences in prevalence of adverse outcomes resulted in over- and underestimation of risk in the Omicron and UK external validation cohorts (Fig 3).

The LMIC-PRIEST score builds on existing clinical triage practices in South Africa, and in addition to parameters used to calculate SATS, the score includes other routinely collected variables. The score is therefore clinically applicable to the intended setting of use. In existing practice, clinicians admitted 14.7% of patients, and discharged patients had a 1% risk of the primary outcome (NPV 0.99, 95% CI 0.99 to 0.99). Use of the score at score thresholds up to <5 could offer gains in sensitivity, but, in the Western Cape, this would increase the proportion of admitted patients with a very small associated reduced risk of false negative triage. Lower thresholds could be used to rapidly and transparently identify a proportion of very low-

**Table 3. Sensitivity, specificity, PPV, NPV and proportion with a positive score at different score thresholds for predicting the primary outcome.**

| | Proportion with score | Sensitivity | Specificity | NPV | PPV |
|---|---|---|---|---|---|
| Development Cohort (Alpha and Delta variants prevalence primary outcome 4.1%) | | | | | |
| >3 | 57.9% | 0.949 (0.945,0.952) | 0.437 (0.435,0.439) | 0.995 (0.995,0.995) | 0.068 (0.067,0.069) |
| >4 | 46.4% | 0.904 (0.899,0.91 | 0.555 (0.553,0.557) | 0.993 (0.992,0.993) | 0.081 (0.079,0.082) |
| >5 | 36.7% | 0.846 (0.84,0.853) | 0.654 (0.652,0.656) | 0.99 (0.99,0.99) | 0.095 (0.094,0.097) |
| >6 | 28.4% | 0.773 (0.765,0.780) | 0.737 (0.736,0.739) | 0.987 (0.986,0.987) | 0.112 (0.11,0.114) |
| >7 | 21.4% | 0.679 (0.671,0.687) | 0.806 (0.804,0.807) | 0.983 (0.983,0.984) | 0.131 (0.128,0.133) |
| Validation Cohort (Omicron variant prevalence primary outcome 2%) | | | | | |
| >2 | 70.7% | 0.962 (0.954,0.969) | 0.298 (0.295,0.3) | 0.997 (0.997,0.998) | 0.027 (0.026,0.028) |
| >3 | 55.3% | 0.929 (0.918,0.938) | 0.454 (0.452,0.457) | 0.997 (0.996,0.997) | 0.033 (0.032,0.035) |
| >4 | 43.1% | 0.841 (0.827,0.854) | 0.577 (0.575,0.58) | 0.994 (0.994,0.995) | 0.039 (0.037,0.04) |
| >5 | 32.9% | 0.766 (0.75,0.782) | 0.68 (0.677,0.682) | 0.993 (0.993,0.994) | 0.046 (0.044,0.048) |
| >6 | 24.6% | 0.663 (0.645,0.68) | 0.763 (0.761,0.765) | 0.991 (0.991,0.992) | 0.054 (0.051,0.056) |
| >7 | 17.6% | 0.555 (0.537,0.574) | 0.831 (0.829,0.833) | 0.989 (0.989,0.99) | 0.063 (0.06,0.066) |
| UK Validation Cohort (Original variant prevalence primary outcome 22.1%) | | | | | |
| >3 | 81.4% | 0.988 (0.984,0.991) | 0.235 (0.229,0.242) | 0.985 (0.981,0.989) | 0.268 (0.262,0.275) |
| >4 | 73.7% | 0.971 (0.966,0.976) | 0.329 (0.322,0.337) | 0.976 (0.971,0.98 | 0.292 (0.284,0.299) |
| >5 | 65.2% | 0.942 (0.935,0.949) | 0.43 (0.422,0.437) | 0.963 (0.958,0.967) | 0.319 (0.311,0.327) |
| >6 | 56.5% | 0.891 (0.881,0.90) | 0.527 (0.52,0.525) | 0.944 (0.94,0.949) | 0.349 (0.34,0.357) |
| >7 | 47.1% | 0.818 (0.806,0.829) | 0.628 (0.62,0.635) | 0.924 (0.919,0.929) | 0.384 (0.375,0.394) |

risk patients who could be discharged from the ED based on information routinely collected at initial assessment.

## Comparison to previous literature

A systematic review found that no risk-stratification scores for patients with suspected COVID had been developed and validated in LMICs [20]. The Nutri-CoV score was subsequently developed and validated in Mexico using data from the first wave of the pandemic and has been proposed for use to triage patient acuity in hospital settings [22]. In internal validation, the score achieved a C-statistic of 0.797 (95% CI 0.765 to 0.829), sensitivity 0.93 (95% CI 0.88 to 0.98), specificity 0.11 (95% CI 0.02 to 0.21) and NPV 0.6 (95% CI 0.34 to 0.87) when estimating death or ICU admission at a recommended threshold. Although not explicitly requiring inpatient investigations, a key predictor in the Nutri-CoV score is diagnosis of pneumonia, which requires either radiological or clinical diagnosis. Consequently, the Nutri-CoV score may not be suitable for rapid identification of low-risk patients suitable for discharge using information available at triage. Unlike the LMIC-PRIEST score, the Nutri-COV score has not undergone external validation in different COVID variants or income settings.

The variables consistently selected by LASSO modelling and used to inform the LMIC-PRIEST score are consistent with other studies [23]. Age, inspired oxygen and oxygen saturations have been found, as in our models, to be highly predictive of adverse outcomes in the ED setting [3, 24]. Although diabetes and heart disease, amongst other comorbidities, have been found to be prognostic in COVID infection, they have not been found to be as highly predictive of adverse outcomes in the ED setting as in our study [3]. However, diabetes has previously been identified as a strong predictor (OR 1.84, 95% CI 1.24 to 2.73) of death in patients with COVID in studies conducted in South Africa [15].

## Strengths and limitations

We followed robust statistical model development techniques that have been rated as low risk of bias and have an adequate sample size for model development [3, 23]. Restricted and unrestricted models derived using different methods for handling missing data, alongside clinical judgement, were used for score development. These measures all reduce the risk of over-optimistic prediction and inclusion of variables by chance that are not truly predictive. The use of cohorts of patients with suspected infection means that the score has been developed and validated in the population of intended use, the population whom ED staff must clinically triage [1]. Our available datasets also comprise multiple COVID waves and different income settings allowing external validation and assessment of generalisability [19].

The cohort used for model development was collected from selected government hospitals using the recently implemented HECTIS system. Use of the HECTIS system, electronic records and linking of data from various sources is in its infancy in this context and is dependent on many data entry points across facilities and institutions. Data collection is not primarily intended for research purposes and may be subject error and missingness. However, the HECTIS system is used clinically to collect and record the physiological and other variables used to calculate SATS in the ED as part of clinical practice [25]. Deaths are recorded if they occurred in, or were notified to, a health facility and deaths which occurred in the community or at other health care facilities and were not notified to the public healthcare system will not be included. Estimated inpatient mortality from COVID during the Omicron wave in the Western Cape was 10.7% [26]. Given only 14.7% of patients in our cohort were admitted for inpatient treatment, we believe our estimated mortality rate of 3.2% is plausible for the Western Cape cohort.

As the UK PRIEST cohort was collected during the first COVID wave (where vaccines were not available) and vaccination/previous infection status was not known in the Western Cape cohort, we could not include vaccination/previous infection status when developing our prognostic model. Our cohort is formed by patients who were tested and diagnosed with COVID and clinical staff performing the initial assessment in the ED had a strong clinical impression of likely infection. This was partly determined by prevalence of infection and clinical guidance, which varied during the study period. Although use of PRIEST study data allowed external validation of our developed score, the intended setting of use is in other LMICs during current waves of the pandemic.

## Implications

During periods of increased COVID prevalence, patients in South Africa with suspected infection were found to bypass primary care and self-present to hospitals [15]. This was associated with excess attendances for patients who required no specific treatment. This partly explains the lower prevalence of the primary outcome in the Western Cape. In the UK, telephone triage was used effectively to reduce ED attendances of lower-risk patients [27]. Disposition decision-making in LMICs is based on clinician experience and gestalt [7]. Existing clinical decision-making was found to perform well with only 14.7% of patients admitted as inpatients and a risk of false negative triage of around 1%. Although clinical-decision rules have been found to rarely out-perform clinician gestalt [28], exercising clinical judgement requires time and experience, which may be limited during periods of increased demand.

Despite, imperfect calibration in external validation, use of the LMIC-PRIEST score at thresholds of three or less would allow identification of very low-risk patients (NPV ≥0.99) across different settings using information routinely collected during initial triage. It also provided better accuracy to generic risk-assessment scores such as NEWS2 and TEWS for patients

with suspected COVID in the Western Cape setting (NEWS C-statistic 0.8 and TEWS C-statistic 0.68 for same primary outcome) [6]. During periods of increased COVID prevalence and corresponding ED attendances, the score could potentially be used by practitioners with basic training to identify very low-risk patients for discharge without full clinical assessment, thereby reducing the risk of hospitals being overwhelmed. A conservative, high sensitivity (0.96) threshold of >2, in the Omicron validation cohort would allow the theoretical identification of 30% of patients as very low-risk and suitable for rapid discharge from the ED using information available following initial triage. Triage score such as LMIC-PREIST are intended to be used in conjunction with clinical judgement and impact studies comparing the use of the LMIC-PRIEST score to existing practice are required to assess the impact of use. Use of a threshold of >5 in the Omicron cohort would achieve a similar sensitivity to clinical decision making to admit patients (0.77 (95% CI 0.76 to 0.78)), but the lower specificity would lead to an increase in proportion of admitted patients from 14.7% to 32.9%.

The wide variation in the prevalence of the primary outcome between settings resulted in miscalibration in external validation. The LMIC-score will need calibration in settings of intended use and this is likely to be easier where there are existing mechanisms for routine data collection such as the HECTIS system. This may simply involve selecting the most appropriate threshold based on the population risk and clinical context. However, the emergence of new variants may require different weightings of predictor variables.

A primary outcome of death and ICU admission/organ support was used to encompass need for hospital admission [3]. However, across all development and validation settings, the score predicted death better than ICU admission/organ support (S9, S10 Tables in S1 File). The accuracy of the LMIC-PRIEST score for the composite outcome should not be used to guide treatment decisions beyond need for admission, such as potential benefit from invasive treatments, as differences in the prediction of death and interventions are likely to mean that the estimation of benefit is inaccurate [29].

## Conclusion

The LMIC-PRIEST score has been developed using robust methods and the score shows generalisable discrimination across a range of COVID variants and income settings. It is specifically designed to be used as part of existing triage practices in South Africa and other LMICs. The score could be used to identify very low-risk patients with suspected COVID infection rapidly and transparently during periods in which health care systems experience increased demand due to a high prevalence of infection. Further external validation may be necessary if the score is used in different settings or novel COVID variants.

## Supporting information

**S1 File. S1 to S18: All supplementary material.**
(DOCX)

## Acknowledgments

This work uses data provided by patients as part of their care and support and the authors wish to recognise the Western Cape Government Health and Wellness (WCGHW) for their contribution of the data that made this research possible, specifically Nesbert Zinyakatira and the team from the Provincial Health Data Centre, Health Impact Assessment Directorate, Western Cape Government Health; and Dr Moosa Parak and the HECTIS team. We further acknowledge and thank the National Health Laboratory Service of South Africa, for their

contribution to the study through the provision of the digitised laboratory results accessed through the Provincial Health Data Centre. Dr Laura Sutton (University of Sheffield) conducted the statistical analysis for the UK PRIEST study and provided a template for the statistical analyses.

## Author Contributions

**Conceptualization:** Carl Marincowitz, Peter Hodkinson, David McAlpine, Steve Goodacre, Peter A. Bath, Yasein Omer, Lee Wallis.

**Data curation:** Peter Hodkinson, David McAlpine, Gordon Fuller, Laura Sbaffi, Madina Hasan, Yasein Omer.

**Formal analysis:** Carl Marincowitz, Gordon Fuller, Steve Goodacre.

**Funding acquisition:** Carl Marincowitz, Peter Hodkinson, Gordon Fuller, Steve Goodacre, Lee Wallis.

**Investigation:** Carl Marincowitz, Peter Hodkinson, Gordon Fuller, Steve Goodacre, Lee Wallis.

**Methodology:** Carl Marincowitz, Peter Hodkinson, David McAlpine, Gordon Fuller, Steve Goodacre, Laura Sbaffi, Lee Wallis.

**Project administration:** Carl Marincowitz.

**Supervision:** Carl Marincowitz, Peter Hodkinson, Peter A. Bath.

**Writing – original draft:** Carl Marincowitz.

**Writing – review & editing:** Carl Marincowitz, Peter Hodkinson, David McAlpine, Gordon Fuller, Steve Goodacre, Peter A. Bath, Laura Sbaffi, Madina Hasan, Yasein Omer, Lee Wallis.

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
