## [Decision Letter · Decision Letter 0]

15 Mar 2023

PONE-D-22-33084LMIC-PRIEST: Derivation and validation of a clinical severity score for acutely ill adults with suspected COVID-19 in a middle-income settingPLOS ONE

Dear Dr. Marincowitz,

Thank you for submitting your manuscript to PLOS ONE. After careful consideration, we feel that it has merit but does not fully meet PLOS ONE’s publication criteria as it currently stands. Therefore, we invite you to submit a revised version of the manuscript that addresses the points raised during the review process.

We look forward to receiving your revised manuscript.

Kind regards,

Christine Kelly, PhD

Academic Editor

PLOS ONE

Journal Requirements:

"CM is a National Institute for Health Research (NIHR) Clinical Lecturer in Emergency Medicine (Grant Number Not Applicable/NA). This work is part of the Grand Challenges ICODA pilot initiative, delivered by Health Data Research UK and funded by the Bill & Melinda Gates Foundation and the Minderoo Foundation. The Provincial Health Data Centre (PHDC), Health Intelligence Directorate, Western Cape Government Health and Wellness acknowledges funding from the United States National Institutes of Health (R01HD080465, U01AI069911), Bill and Melinda Gates Foundation (1164272; 1191327; INV-004657, INV-017293), the Wellcome Trust (203135/Z/16/Z), the United States Agency for International Development (72067418CA00023)."

Reviewers' comments:

Reviewer's Responses to Questions

**Comments to the Author**

1. Is the manuscript technically sound, and do the data support the conclusions?

Reviewer #1: Yes

2. Has the statistical analysis been performed appropriately and rigorously? 

Reviewer #1: Yes

3. Have the authors made all data underlying the findings in their manuscript fully available?

Reviewer #1: Yes

4. Is the manuscript presented in an intelligible fashion and written in standard English?

Reviewer #1: Yes

5. Review Comments to the Author

Reviewer #1: OVERALL

Many thanks for asking me to review this paper which I enjoyed reading. The authors describe the derivation and validation of a novel risk prediction tool for patients with suspected COVID for use in LMICs. The authors have developed the tool using robust methodology in accordance with TRIPOD recommendations. I think that this paper would make a useful contribution to the literature and would recommend it is published with a few minor amendments. I have also offered a few suggestions for additional analyses that may be of interest.

GENERAL COMMENTS TO BE ADDRESSED

1. UNCERTAINTY ABOUT COMPLETENESS OF OUTCOME DATA - As to be expected when using routinely-collected clinical data there are high rates of missingness for some predictor parameters of interest, although the authors have assessed different analysis and imputation approaches to explore this. Can the authors offer some assurance regarding the completeness of outcome ascertainment? For example, if a patient was initially discharged at index attendance and then died at home or represented to (either the same or another) hospital and was admitted to ICU, would this be reliably captured? Can the authors provide any references for completeness of outcome capture with the HECTIS system or if uncertain acknowledge this as a limitation.

2. EFFECT OF VACCINATION STATUS OR PRIOR INFECTION ON OUTCOME - The authors have discussed changes in outcome over time and between settings and relate this to emergence of new variants. Vaccination status clearly also has an impact on outcome. Do the authors have any data available on vaccination status of the cohorts, its impact on outcome and prognostic performance of LMIC-PRIEST? If not, I would suggest that this is addressed in the discussion.

3. OXYGEN SATURATIONS AND USE OF SUPPLEMENTAL OXYGEN - Could the authors please clarify whether the oxygen saturations used to develop the model are measured off supplemental oxygen. Intuitively, I would expect supplemental oxygen use and baseline saturations to be collinear so not both required for prognostic model. Or does this relate to the fact that model might be used in low resources settings where patients with low oxygen saturations might not necessarily receive oxygen?

4. CALIBRATION FOR USE IN SUBSEQUENT WAVES AND NEW SETTINGS - The authors comment that the LMIC-score will need calibration for the intended setting of use. Could they offer some thoughts in the discussion on how this might be achieved efficiently in a resource limited setting?

ADDITIONAL CONSIDERATIONS

1. COVID-SPECIFIC vs. GENERIC ADVERSE OUTCOME PREDICTION - To what extent do COVID patients require a disease specific risk assessment tool? In this respect, I think it would be interesting to compare the prognostic performance of LMIC-PRIEST to generic risk assessment tools (e.g. TEWS, NEWS2 etc)? There would be challenges to implementing a new risk assessment in clinical practice, so would be good to demonstrate that LMIC-PRIEST adds value.

2. VARIATION IN PERFORMANCE BY CONFIRMED COVID STATUS - I appreciate that the authors aimed to develop a risk assessment tool that could be applied for use in patients in emergency departments for whom COVID test results are not known and so have appropriately included patients with suspected COVID in their derivation population. In the initial derivation cohort, COVID PCR positivity appears associated with adverse outcome (Table 1). It is conceivable that lateral flow tests may be available in some LMIC settings, so I think it would be interesting to undertake exploratory analyses to look at LMIC-PRIEST prognostic performance in patients with confirmed vs non-confirmed COVID.

3. VARIATION IN PERFORMANCE WITH COVID PREVALENCE - Related to (2.), the authors comments that the 'clinical impression of likely [COVID] infection…. was partly determined by prevalence of infection…. that varied during the study period.' The authors discuss calibration for variant dependent changes in outcomes, but it would be interesting to understand the impact on accuracy of changes in COVID incidence. During COVID waves, the pre-test probability of COVID for patients with a compatible presentations can change rapidly. What would the impact on prognostic accuracy if confirmed COVID prevalence in suspected COVID cohort was 5% or 50% rather than 24%.

4. COMPARATIVE IMPACT ON PATIENT DISPOSITION OF FORMAL RISK ASSESSMENT vs. CLINICAL JUDGEMENT - In the 2nd paragraph of the discussion, the authors describe prognostic accuracy of initial clinical disposition decision and discuss impact on admission rate and false negative triage of instead using LMIC-PRIEST to guide disposition. I think that this is particularly interesting and would suggest describing in the results the hypothetical impact on patient disposition of applying LMIC-PRIEST at different thresholds.

MINOR TYPOGRAPHICAL ERRORS

1. Comma misplaced in first sentence of results in abstract - 'We analysed 305,564, derivation.....'

2. Explain acronym TEWS at first use

3. Table 3 - I would customarily show results of development cohort in the top rows.

6. PLOS authors have the option to publish the peer review history of their article (what does this mean?). If published, this will include your full peer review and any attached files.

Reviewer #1: **Yes: **Dr Stephen Aston

---

## [Author Response · Author response to Decision Letter 0]

25 May 2023

Dear Dr Kelly,

Thank you for the useful and detailed comments and the opportunity to further revise and resubmit our manuscript to PLOS ONE (PONE-D-22-33084). 

We hope these revisions and responses meet with your approval and those of the reviewers. 

Yours Sincerely,

Carl Marincowitz

We have addressed the comments and suggestions, revising our manuscript as detailed below:

Response: We have amended the manuscript so that it now complies with these templates.

Response: Details regarding anonymisation of data and waiving of the need for patient consent by relevant ethics committees/other institutions are provided in the final sentence of the first paragraph of the ethics section and final sentences of the second paragraph of the ethics section.

Response: This has been amended.

"CM is a National Institute for Health Research (NIHR) Clinical Lecturer in Emergency Medicine (Grant Number Not Applicable/NA). This work is part of the Grand Challenges ICODA pilot initiative, delivered by Health Data Research UK and funded by the Bill & Melinda Gates Foundation and the Minderoo Foundation. The Provincial Health Data Centre (PHDC), Health Intelligence Directorate, Western Cape Government Health and Wellness acknowledges funding from the United States National Institutes of Health (R01HD080465, U01AI069911), Bill and Melinda Gates Foundation (1164272; 1191327; INV-004657, INV-017293), the Wellcome Trust (203135/Z/16/Z), the United States Agency for International Development (72067418CA00023)."

Response: The funders has not role in the study. We have added the statement: "The funders had no role in study design, data collection and analysis, decision to publish, or preparation of the manuscript." to the end of the funding statement in our manuscript. We will include a copy of our full funding statement in our cover letter and is also included below.

CM is a National Institute for Health Research (NIHR) Clinical Lecturer in Emergency Medicine (Grant Number Not Applicable/NA). This work is part of the Grand Challenges ICODA pilot initiative, delivered by Health Data Research UK and funded by the Bill & Melinda Gates Foundation and the Minderoo Foundation. The Provincial Health Data Centre (PHDC), Health Intelligence Directorate, Western Cape Government Health and Wellness acknowledges funding from the United States National Institutes of Health (R01HD080465, U01AI069911), Bill and Melinda Gates Foundation (1164272; 1191327; INV-004657, INV-017293), the Wellcome Trust (203135/Z/16/Z), the United States Agency for International Development (72067418CA00023). The funders had no role in study design, data collection and analysis, decision to publish, or preparation of the manuscript.

Response: Captions for supporting materials are provided at the end of the manuscript and in-text citations have also been amended. 

Response: The reference list has been reviewed.

Reviewer #1: OVERALL

Many thanks for asking me to review this paper which I enjoyed reading. The authors describe the derivation and validation of a novel risk prediction tool for patients with suspected COVID for use in LMICs. The authors have developed the tool using robust methodology in accordance with TRIPOD recommendations. I think that this paper would make a useful contribution to the literature and would recommend it is published with a few minor amendments. I have also offered a few suggestions for additional analyses that may be of interest.

GENERAL COMMENTS TO BE ADDRESSED

1. UNCERTAINTY ABOUT COMPLETENESS OF OUTCOME DATA - As to be expected when using routinely-collected clinical data there are high rates of missingness for some predictor parameters of interest, although the authors have assessed different analysis and imputation approaches to explore this. Can the authors offer some assurance regarding the completeness of outcome ascertainment? For example, if a patient was initially discharged at index attendance and then died at home or represented to (either the same or another) hospital and was admitted to ICU, would this be reliably captured? Can the authors provide any references for completeness of outcome capture with the HECTIS system or if uncertain acknowledge this as a limitation.

Response:

The routinely collected linked health care datasets provided by the Western Cape Department of Health and Wellness will reliably include all deaths and admission outcomes (such as ICU admission) which occurred in health facilities, or were notified by community practitioners, to the participating public hospitals. Since the introduction of the routinely collected electronic data repositories recording of such deaths within the study dataset occurs as part of routine clinical practice. We will have reliably captured all deaths following discharge for patient who re-attended to participating hospitals (it is unlikely they would have re-attended else-where) and deaths where patients were registered with community practitioners linked to the public health care system. 

Our mortality estimates are plausible in the context of other studies. Some 76.6% of the study population attended the ED after March 2021 (and therefore after the first and second COVID waves) and 140,520/446,084 (31.5%) presented during the Omicron period (November 2021 onwards). Inpatient case fatality rates in South Africa for patients with confirmed COVID had fallen from a high of 28.8% (second wave) to 21.5% during the third wave (April to November 2021) and were estimated to be 10·7% during the Omicron Period.1 Given only 14.7% of patients in our cohort were admitted for inpatient treatment, we believe our estimated mortality rate of 3.2% is plausible.

We have added 5th, 6th and 7th sentences of the second paragraph of the strenghts and limitations section to highlight which death might not be included in estimating our outcomes and the comparison to estimated mortality for inpatients.

1. Jassat W, Abdool Karim SS, Mudara C, et al. Clinical severity of COVID-19 in patients admitted to hospital during the omicron wave in South Africa: a retrospective observational study. The Lancet Global Health. 2022;10:e961-e969.

2. EFFECT OF VACCINATION STATUS OR PRIOR INFECTION ON OUTCOME - The authors have discussed changes in outcome over time and between settings and relate this to emergence of new variants. Vaccination status clearly also has an impact on outcome. Do the authors have any data available on vaccination status of the cohorts, its impact on outcome and prognostic performance of LMIC-PRIEST? If not, I would suggest that this is addressed in the discussion.

Response: As the UK PRIEST cohort was collected during the first wave, no patients were vaccinated and are unlikely to have been previously infected.

We were unable to gather information retrospectively on vaccination status/previous infection of patients in the Western Cape cohort, and although we agree that this would have been ideal, vaccinations in South Africa were only rolled out to the public from May 2021, with a phased roll out starting with elderly and high-risk patients. There was a relatively slow uptake of vaccination for various reasons in South Africa, with only around 20% fully vaccinated by the time the Omicron variant hit (December 2021) and the impact of vaccination was unclear.

We now include the first sentence of the third paragraph of the strengths and limitations section to highlight this.

3. OXYGEN SATURATIONS AND USE OF SUPPLEMENTAL OXYGEN - Could the authors please clarify whether the oxygen saturations used to develop the model are measured off supplemental oxygen. Intuitively, I would expect supplemental oxygen use and baseline saturations to be collinear so not both required for prognostic model. Or does this relate to the fact that model might be used in low resources settings where patients with low oxygen saturations might not necessarily receive oxygen?

Response: Oxygen saturations include those measured where supplemental oxygen was already being administered when patients were initially triaged in the ED. Our decision to include this measure of saturation was pragmatic as it reflects use of oxygen saturations when included in the NEWS2 early warning score and our experience in developing the UK PRIEST score. Initially, a measure of oxygen saturation to inspired oxygen concentration ratio was used in modelling to develop the PRIEST score, however feedback from clinicians indicated this would not be feasible to use practically.

4. CALIBRATION FOR USE IN SUBSEQUENT WAVES AND NEW SETTINGS - The authors comment that the LMIC-score will need calibration for the intended setting of use. Could they offer some thoughts in the discussion on how this might be achieved efficiently in a resource limited setting?

Response: We have added to the 3rd paragraph of the implications section to highlight that this is likely to be easier in setting with existing mechanisms for routine data collection.

ADDITIONAL CONSIDERATIONS

1. COVID-SPECIFIC vs. GENERIC ADVERSE OUTCOME PREDICTION - To what extent do COVID patients require a disease specific risk assessment tool? In this respect, I think it would be interesting to compare the prognostic performance of LMIC-PRIEST to generic risk assessment tools (e.g. TEWS, NEWS2 etc)? There would be challenges to implementing a new risk assessment in clinical practice, so would be good to demonstrate that LMIC-PRIEST adds value.

Response: We now include the second sentence of the second paragraph of the implications section to highlight that the LMIC-PRIEST score has higher estimated discrimination than NEWS2 and TEWS in the Western Cape setting for patients with suspected COVID alongside a reference to our study validating these and other risk scores in the Western Cape cohort of patients

2. VARIATION IN PERFORMANCE BY CONFIRMED COVID STATUS - I appreciate that the authors aimed to develop a risk assessment tool that could be applied for use in patients in emergency departments for whom COVID test results are not known and so have appropriately included patients with suspected COVID in their derivation population. In the initial derivation cohort, COVID PCR positivity appears associated with adverse outcome (Table 1). It is conceivable that lateral flow tests may be available in some LMIC settings, so I think it would be interesting to undertake exploratory analyses to look at LMIC-PRIEST prognostic performance in patients with confirmed vs non-confirmed COVID.

Response: We agree this may this interesting, however do not have the capacity to perform additional analysis as the funding and allotted time for this study has now been used.

Additionally, due to different testing protocols which occurred within the study period and that was implemented in different settings, COVID testing was used in a variable way in the cohorts (in PRIEST it was only done for admitted patients). We also don't have information to determine whether testing was negative or not done. This means that the results of such analysis would need to be presented with significant caveats and could be open to misinterpretation.

We have deliberately used a cohort of patients with a clinical suspicion of suspected COVID to reflect the population we intend the developed tool to help triage and reflects the population where rapid triage of patients in the urgent and emergency care setting may be needed. Many other studies, such as ISARIC-4C have developed robust prognostic models for patients with confirmed COVID and include additional investigations such as blood tests. We wanted to develop a prognostic model and tool using only variables available at initial triage in the urgent and emergency care setting.

3. VARIATION IN PERFORMANCE WITH COVID PREVALENCE - Related to (2.), the authors comments that the 'clinical impression of likely [COVID] infection…. was partly determined by prevalence of infection…. that varied during the study period.' The authors discuss calibration for variant dependent changes in outcomes, but it would be interesting to understand the impact on accuracy of changes in COVID incidence. During COVID waves, the pre-test probability of COVID for patients with a compatible presentations can change rapidly. What would the impact on prognostic accuracy if confirmed COVID prevalence in suspected COVID cohort was 5% or 50% rather than 24%.

Response: Unfortunately, we do not think it is possible to answer this with our available data and confirmation of COVID prevalence will be determined as much by testing protocols and true COVID prevalence. As outlined in our response to point 2, , COVID testing was used in a variable way in the cohorts (in PRIEST it was only done for admitted patients). We also don't have information to determine whether testing was negative or not done. This means that the results of such analysis would need to be presented with significant caveats and could be open to misinterpretation.

4. COMPARATIVE IMPACT ON PATIENT DISPOSITION OF FORMAL RISK ASSESSMENT vs. CLINICAL JUDGEMENT - In the 2nd paragraph of the discussion, the authors describe prognostic accuracy of initial clinical disposition decision and discuss impact on admission rate and false negative triage of instead using LMIC-PRIEST to guide disposition. I think that this is particularly interesting and would suggest describing in the results the hypothetical impact on patient disposition of applying LMIC-PRIEST at different thresholds.

Response: We now include a discussion of use of LMIC-PRIEST score at different thresholds in comparison to clinical decision making at the end of the second paragraph of the implications section. We also highlight that triage score such as LMIC-PREIST are intended to be used in conjunction with clinical judgement and impact studies comparing the use of the LMIC-PRIEST score to existing practice are required to assess the impact of use. We also envision LMIC-PRIEST to be used to rapidly identify a low-risk population of patients who could be discharged from the ED following triage as opposed to solely guiding need for inpatient admission. 

MINOR TYPOGRAPHICAL ERRORS

1. Comma misplaced in first sentence of results in abstract - 'We analysed 305,564, derivation.....'

Response: Thank you for spotting this, it now corrected.

2. Explain acronym TEWS at first use

Response: This has been provided.

3. Table 3 - I would customarily show results of development cohort in the top rows.

Response: The table has been amended as suggested.

---

## [Editor Report · Decision Letter 1]

31 May 2023

LMIC-PRIEST: Derivation and validation of a clinical severity score for acutely ill adults with suspected COVID-19 in a middle-income setting

PONE-D-22-33084R1

Dear Dr. Marincowitz,

We’re pleased to inform you that your manuscript has been judged scientifically suitable for publication and will be formally accepted for publication once it meets all outstanding technical requirements.

Kind regards,

Christine Kelly, PhD

Academic Editor

PLOS ONE
---

## [Editor Report · Acceptance letter]

5 Jun 2023

PONE-D-22-33084R1 

LMIC-PRIEST: Derivation and validation of a clinical severity score for acutely ill adults with suspected COVID-19 in a middle-income setting 

Dear Dr. Marincowitz:

I'm pleased to inform you that your manuscript has been deemed suitable for publication in PLOS ONE. Congratulations! Your manuscript is now with our production department. 

Kind regards, 

on behalf of

Dr. Christine Kelly 

Academic Editor

PLOS ONE